# Diffusion MRI GAN synthesizing fibre orientation distribution data using generative adversarial networks
Sebastian Vellmer [1,2] ✉, Dogu Baran Aydogan [3,4], Timo Roine [4], Alberto Cacciola [5], Thomas Picht[1,2] & Lucius S. Fekonja [1,2] ✉

Machine learning may enhance clinical data analysis but requires large amounts of training data, which are scarce for rare pathologies. While generative neural network models can create realistic synthetic data such as 3D MRI volumes and, thus, augment training datasets, the generation of complex data remains challenging. Fibre orientation distributions (FODs) represent one such complex data type, modelling diffusion as spherical harmonics with stored weights as multiple three-dimensional volumes. We successfully trained an α-WGAN combining a generative adversarial network and a variational autoencoder to generate synthetic FODs, using the Human Connectome Project (HCP) data. Our resulting synthetic FODs produce anatomically accurate fibre bundles and connectomes, with properties matching those from our validation dataset. Our approach extends beyond FODs and could be adapted for generating various types of complex medical imaging data, particularly valuable for augmenting limited clinical datasets.

In the past decade, deep learning techniques have shown great potential for computer vision tasks, which are particularly interesting for the automatised analysis of clinical imaging data. Among many other applications, Convolutional Neural Networks (CNN) have shown remarkable performance in clinical tasks such as lesion detection, classification and segmentation of MRI data[1–3]. However, the success of neural network training essentially depends on the availability of a sufficient amount of training data[4]. Since the collection of clinical data is a time-consuming and expensive procedure and the use of personal data is often restricted, the amount of training data for medical purposes is in many cases very limited, particularly in the case of rare pathologies. In order to improve the performance and thereby to extend the applicability of deep learning methods in medicine, training data sets may be augmented by synthetic data generated by neural networks[5–7].

One unsupervised training method for CNNs that generate synthetic images is given by generative adversarial networks (GANs)[8]. In GANs, the training is implemented as the competition between two CNNs. The Generator transforms noise into an image, while the Discriminator classifies these images as either generated (fake) or part of the training data (real). After successful training, the Generator creates data that are indistinguishable from the training for the Discriminator and the

human eye. The resulting CNN may produce realistic images that are, nevertheless, different from the original training set. It has already been shown that it is possible to generate realistic 2D MRI slices from random numbers[9,10] or entire 3D volumes[11]. These methods can generate images that are different from the training data by learning the data distribution itself, and thus may augment the variability of a data set. GANs have even been used for complex 4D MRI data, for instance, to generate MRI time series of the heart where its motion is prescribed by a mathematical model[12] or to complete undersampled data for a time series of liver MRI[13]. Another interesting application of GANs is spatially dependent diffusion MRI data being highly valuable in network neuroscience[14] and, particularly, in neurosurgical planning[15]. Deducible from diffusion data, the connectome has been of most importance in neuroscience since the introduction of this term[16] and refers to a comprehensive map of neural connections in the brain, however, its potential with regard to clinical diagnostics has not been exploited so far[17] due to its demand on resources that can be reduced drastically applying machine learning techniques in data acquisition and analysis. The structural connectome can be computed based on diffusion MRI, which can provide detailed information of the white matter (WM) fibre pathways non-invasively. In diffusion MRI, the random motion of water molecules attenuates the MRI signal and

[1]Department of Neurosurgery, Charité Universitätsmedizin Berlin, Berlin, Germany. [2]Cluster of Excellence, Matters of Activity, Image Space Material, Berlin, Germany. [3]A.I. Virtanen Institute for Molecular Sciences, University of Eastern Finland, Kuopio, Finland. [4]Department of Neuroscience and Biomedical Engineering, Aalto University School of Science, Espoo, Finland. [5]Brain Mapping Lab, Department of Biomedical, Dental Sciences and Morphological and Functional Images, University of Messina, Messina, Italy. ✉e-mail: sebastian.vellmer@charite.de; lucius.fekonja@charite.de

**Fig. 1 | Generated FODs.** The figure illustrates nine different, synthetic FOD sets in columns 1–3 in comparison to one validation FOD set (column 4) in coronal, axial and saggital views. FOD isosurfaces are shown in each voxel with the colour encodings: blue (top-bottom), green (front-back) and red (left-right). For orientation, we plotted a grayscale map of the zeroth FOD coefficient in the background. The coronal view shows a section through the thalamus and the pons. We find a similar contrast for generated and validation FODs, especially for the pons, corpus callossum, the corticospinal tract and even in the difficult crossing fibre regions in centrum semiovale (see zoom). The axial view depicts a section through the striate body, the thalamus and the internal capsule, with a zoom highlighting the white matter area around the optic radiation. The sagittal slice demonstrates a section through the insula, and a zoom to the posterior part of the insula and Wernicke's speech region.

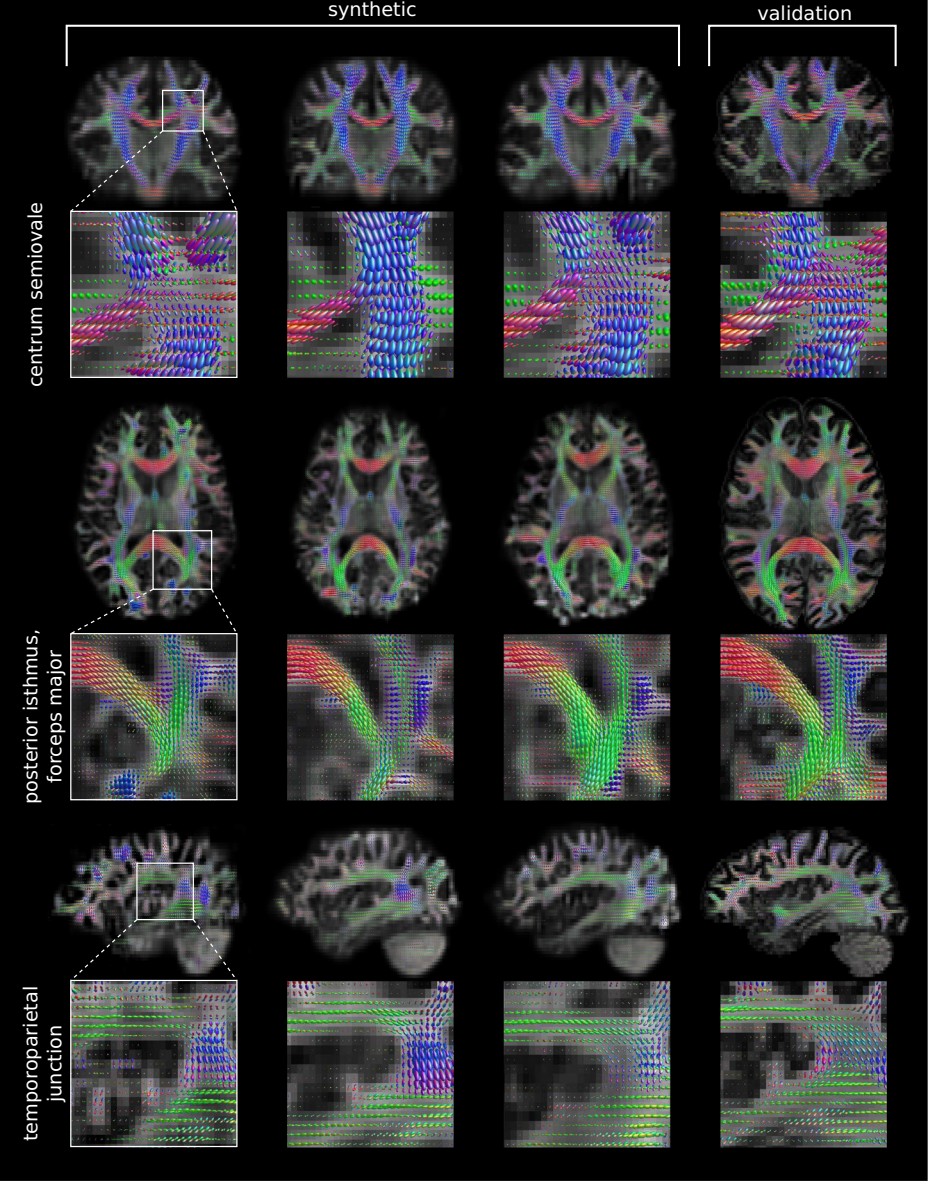

allows the modelling the local fibre architecture and microstructure. Techniques, such as constrained spherical deconvolution (CSD)[18,19] permit to assess complex WM organisation, overcoming the inherent limitations of the diffusion tensor model in crossing fibres regions, which constitute up to 90% of white matter voxels[20]. With high-quality estimates of the FODs, subsequent tractography algorithms can reconstruct structural brain connections and measure apparent fibre density, for example, in clinical studies[21]. Both, the time-consuming acquisition of diffusion MRI data and the computationally expensive analysis required to obtain tractograms or connectomes could be facilitated adapting deep learning techniques[22]. To achieve a clinically sufficient performance of such applications, a large amount of training data is required that in turn may be augmented by synthetic data.

Here, we generalised a generative model that successfully generates 3D MRI volumes of the brain from random vectors[23] to 4D diffusion MRI data, the FODs. We adapted the structure of $\alpha$-WGAN that avoids mode collapse problems and blurry images by introducing an additional variational auto-encoder to a GAN[24]. Furthermore, the Wasserstein loss function with Gradient Penalty (WGAN-GP)[25] is used to prevent unstable training. The proposed GAN model successfully generates FODs as 4D images from

vectors of Gaussian noise, which is a major conceptual advancement in contrast to generating 3D data due to the complex dependencies present in the 4th dimension of diffusion MRI data. We demonstrate the usability of the model and, more importantly, the quality and anatomical validity of the generated 4D image datasets by comparing the connectomes and single fibre tracts derived from generated FODs with those obtained from a validation dataset. For the connectome comparisson, we use a complex network measure of brain connectivity, the global efficiency[26,27] and the distributions of mantel correlations[28] of connectome pairs. For the single tracts, we present the dice-scores distributions, tract volumes and the fibre densities along the tracts. The generated data is highly useful for the augmentation of training sets in which machine learning is applied to perform on FODs. More importantly, our work demonstrates that $\alpha$-WGAN can be generalised to augment various 4D MRI sets, which would be useful especially for pathological data where the amount of data is more limited.

## Results

In total, we generated 100 synthetic FOD data sets. As presented in Figs. 1–4, our trained Generator may produce FODs with anatomically meaningful structures that resemble the training data. We did not observe mode collapse

**Fig. 2 | Synthetic FOD volumes in axial view.** The figure shows axial slices of single volumes representing coefficient maps from the harmonic spherical deconvolution. In the first five rows, we present synthetic data sets and in the sixth row we show one FOD from the validation set for comparison. The number of the presented volume in each column refers to its position in the output volume of the MRtrix3 algorithm.

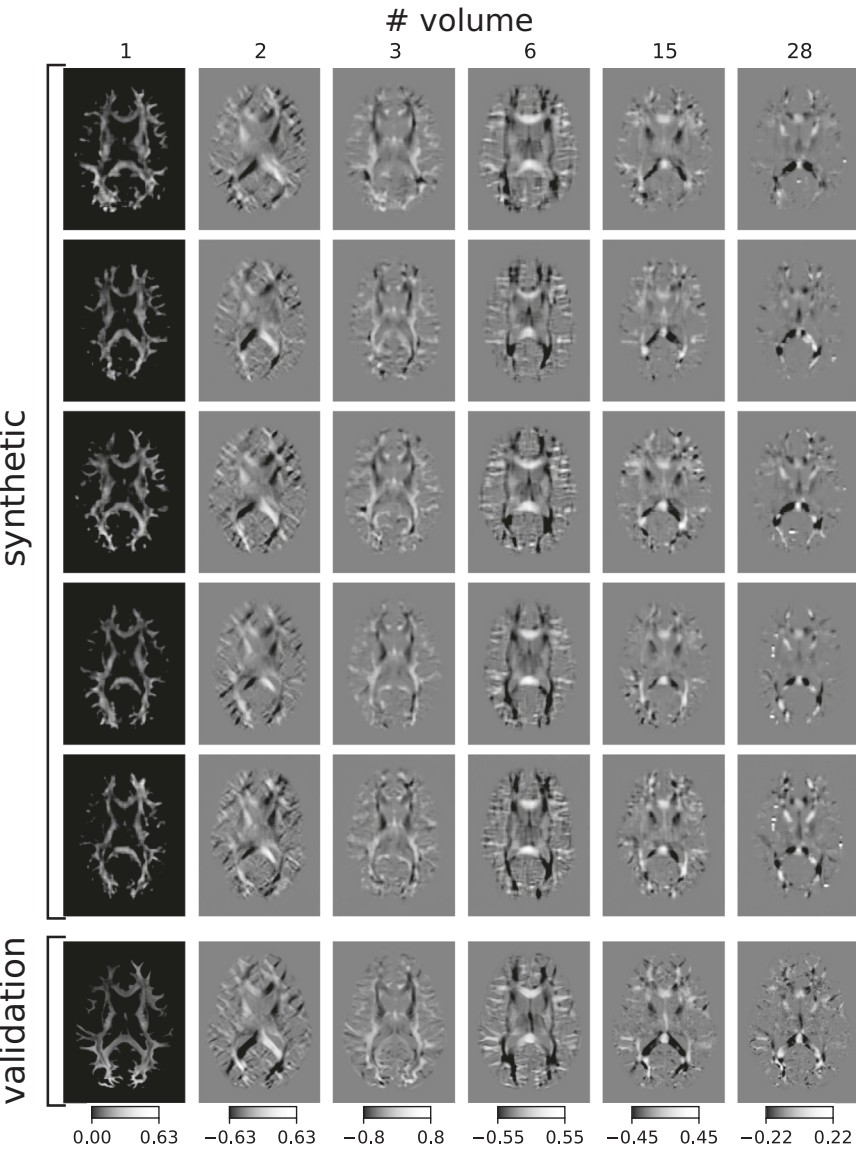

in the data after we drastically increased the contribution of the $L_1$ norm in the loss function Eq. (1), shifting our model more towards a VAE.

### Visual inspection and direct analysis of FOD values

In Fig. 1 we show nine different images of randomly selected synthetic FODs, three for each plane in the first three columns and one FOD from the validation set for comparison in the fourth column. For the visualisation, we used MRview from the MRtrix3 software package[29,30]. In the background, we show the respective layer of the first FOD volume for a better orientation. In all FODs, it was possible to identify the main anatomical structure of the brain such as the ventricles, basal ganglia, corpus callosum (CC), the cerebellum etc. The synthetic data also captured characteristic white matter tracts in the FOD structures, for which we present three examples in Fig. 1. As the first example, we present the centrum semiovale in the coronal section, in which fibres from the corticospinal tract that are predominantly aligned caudocranially (blue colour encoding) cross fibres from the CC that are aligned from left to right (red colour encoding). As a second example, we present the region of the forceps major in the axial section, which connects the occipital parts of the hemispheres and merges with the optic radiation that is mainly aligned in anterior-posterior direction (green colour encoding). As the last example, we chose the fasiculus arcuatus in the sagittal section that connects the Broca's and Wernicke's areas.

In Figs. 2–3, we present axial, coronal and sagittal maps of coefficients of the spherical harmonic functions representing the FODs. The single volumes of the synthetic data show similar contrast in anatomical areas compared to validation data. However, the synthetic data contained few voxels with unphysiologically high or low values, as can be seen for instance in the 28th volume in the fourth row in Fig. 3 and, compared to the validation data, the synthetic data are blurred, a known issue of VAEs. In some FODs, we also see a few unplausible circular structures or sometimes spots of protruding intensities near the cortex which may occur due to a high inter-individual variability of the training data in those areas. We mostly find overlapping histograms for the voxel values presented in Fig. 4B. Differences can be found in the first volume, in which our synthetic data has also some negative values in contrast to the validation data that is exclusively positive. For some volumes, the range of values is larger for the synthetic data. However, considering the logarithmic scale only few voxels may have unplausible high values. Comparing the pairwise squared differences, we find $(18.4 \pm 6.0) \times 10^3$ for the synthetic set and $(28.1 \pm 4.5) \times 10^3$ for the validation set indicating that the variation is lower in the synthetic data.

### Tractography and structural connectomes

For all generated data, we were able to process the synthesised FODs and to generate meaningful tractography-derived connectivity matrices, resulting

**Fig. 3 | Visual comparison of synthetic and validation FODs. A** Synthetic FOD volumes in coronal view. See caption of Fig. 2 for details. **B** Histograms of FODs. Distributions of FOD voxels for the entire synthetic data set corresponding to the volumes labelled above in A. Synthetic voxel data in red, validation voxel data in blue. Note the logarithmic scale.

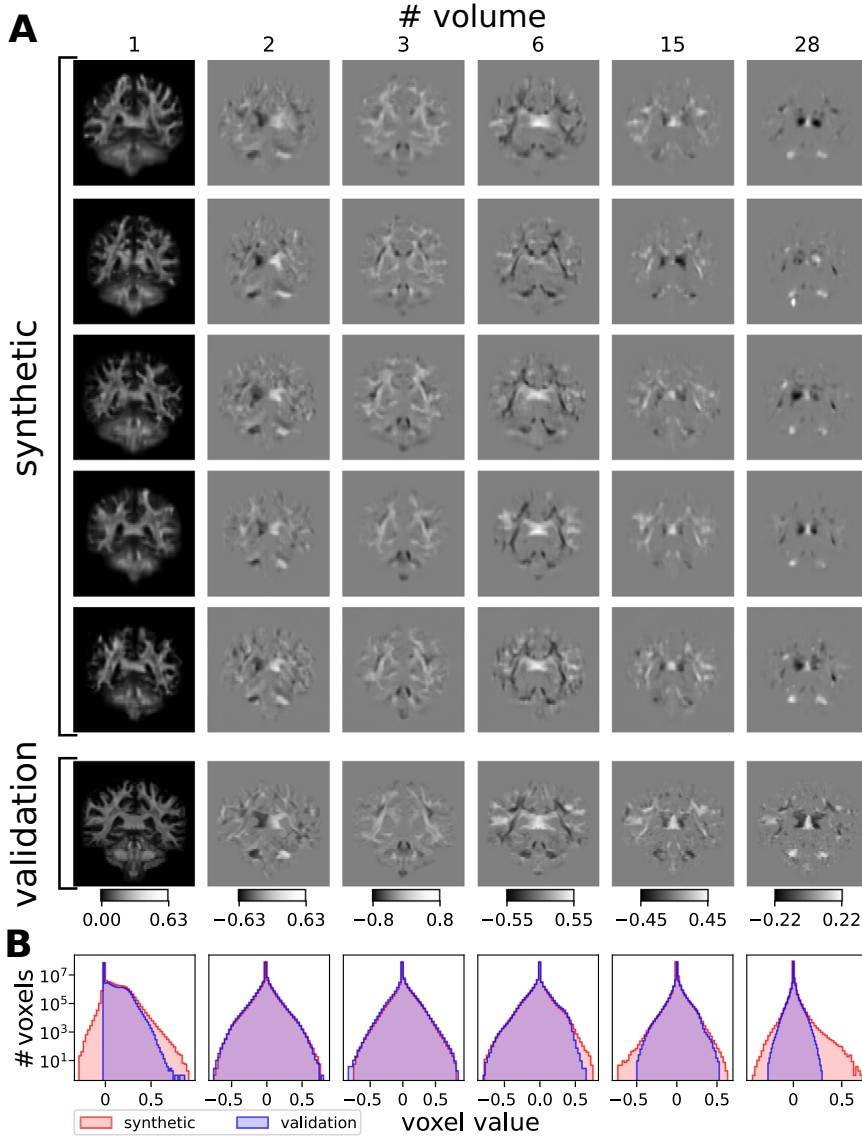

in structural connectomes as shown in Fig. 5. We compare these connectomes with connectomes derived from the validation data in Fig. 6. By visual inspection, we find similar structures in connectomes of both groups, for instance, dense blocks in the corners and in the middle that are interrupted by blocks with lower densities (cf. Fig. 6A). This similarity of the connectomes can also be visualised by the histograms of all connectome elements presented in panel Fig. 6B. We find small bias in the connectome elements of synthetic data compared to the validation data that is caused by an increased streamline weighting from the SIFT2 algorithm of 8.6% on average. To compensate for that, we normalise the connectomes such that the sum of elements of each individual connectome is two, considering that each connection appears two times in the symmetric connectome. For a quantitative comparison, we calculated the global efficiency as an example of a complex network measure for each normalised connectome and show the resulting histograms in Fig. 6C. We find that the validation data exhibit slightly larger global efficiency and a broader distribution with mean and standard deviation of $(6.59 \pm 0.22) \times 10^{-4}$ compared to the synthetic data with $(6.47 \pm 0.16) \times 10^{-4}$. The small relative difference in global efficiency of 1% is, however, statistically significant with a $p$-value of $3.3 \times 10^{-7}$ calculated by the Wilcoxon-Mann-Whitney test implemented in Python SciPy version 1.11.1. Furthermore, we performed the Mantel test to determine correlations between the connectomes. In contrast to the analysis of the global

efficiencies, we obtain similar distributions of correlation coefficients for connectomes within the generated data and generated vs. validation data (see Fig. 6C) indicating similarity of generated and validation connectomes. The distribution within the validation data is more narrow and shifted towards larger coefficients.

### Individual fibre tracts

We reconstructed single fibre tracts with the deterministic Peak-FOD based FACT, and the probabilistic iFOD2 algorithms. We obtained anatomically meaningful tractograms derived from the synthetic FODs, for which we present one example containing the CC, the arcuate fascicle (AF) and the corticospinal tract (top row) and the inferior fronto-occipital fascicle, the optic radiation and the uncinate fascicle (bottom row) derived by the FACT algorithm, cf. Fig. 7A. The selection of bundles includes association, commissural and projection fibres. For a quantitative analysis of our data, we compared the volumes of the reconstructed tracts of the synthetic and the validation data by the dice score (see Eq. (4)). The resulting dice-score distributions for the tracts CC, left and right AF and left and right corticospinal tract (CST) are presented in Fig. 7B. The distributions of synthetic vs. validation data are similar in shape and location to the distributions of validation vs. validation data showing that synthetic tracts are similar to validation tracts in shape and size. For synthetic vs. synthetic data, the

**Fig. 4 | Synthetic FOD volumes in sagittal view.**
See caption of Fig. 2 for details.

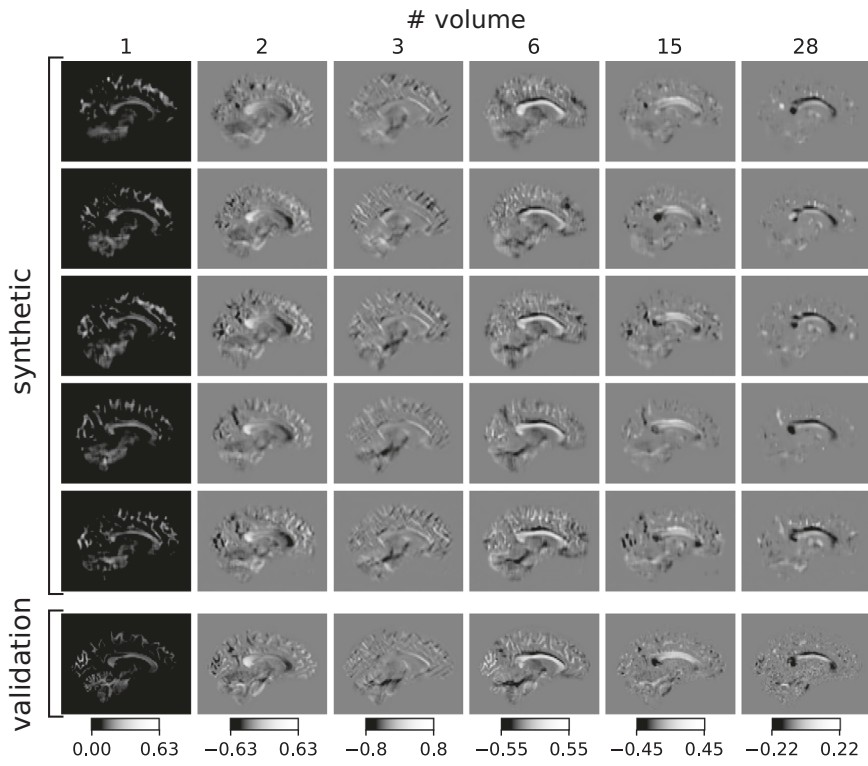

distributions are similar in shape but slightly shifted towards higher values. This finding indicates that the variation of the tracts' shapes is lower. We do not find high dice scores close to one such that we may exclude a mode collapse problem. However, in two of the generated FODs, small fibre tracts such as the UF could not be reconstructed even with the iFOD2 algorithm. In both cases, the UF fibre-bundle mask derived with tractSeg was discontinuous. Therefore, we find no overlap indicated by a zero or low dice-score, respectively, for a few synthetic vs. synthetic and synthetic vs. validation data pairs but not for validation vs. validation pairs.

For the iFOD2-derived tracts, the mean volumes are presented in Fig. 8A, beginning with the smallest tract on the left-hand side to the largest one on the right-hand side. For all tracts, we find that synthetic data yielded smaller tracts in terms of volume compared to the validation data. The ratios of the synthetic and validation's mean volumes from synthetic and validation data are presented in Fig. 8B. Over all tracts, we find that the synthetic ones are on average $8 \pm 4$ % smaller than the ones from the validation set. As a further analysis, the apparent fibre densities along the AF, IFOF, OR, UF and the CST calculated with the iFOD2-derived tracts are shown in Fig. 8C. Synthetic and validation tracts are characterised by a similar progression of the fibre densities with peaks and dips at the same locations indicating that the generated FODs resemble the validation data set. We only find smaller deviations in the absolute values, especially for the small UF.

The relative differences of fibre-density mean and standard deviations (see Eq. (5)) in the AAL parcels reveal a bias in the cerebellum where they are up to 12 and 45 percent, respectively. For other regions, we find lower values under 5 and 20 percent, respectively, that are distributed along the cortex without deducible bias towards a certain area. The highly individual structure of the cerebellum may be more difficult to be learned by our model.

## Discussion

In recent years, generative models have demonstrated versatile applications in many studies devoted to medical imaging data. For instance, GANs have been used to convert between different imaging modalities, such as T2 MRI to CT contrasts[31], or to generate synthetic T1 MRI out of Gaussian noise[32]. Here we have shown that GANs are capable of generating FODs as an example of complex MRI data that consist of several volumes.

We presented an adapted version of the $\alpha$-Wasserstein GAN[23] that combines the Wasserstein GAN with a VAE. For training, we derived 965 FODs from the HCP dataset and performed aligned rotations to augment the trainings set to 4825 subjects. We trained our model with FODs derived from the preprocessed HCP data set[33] with a spatial resolution of $128 \times 128 \times 64$ and $l_{max} = 6$ corresponding to 28 volumes of spherical harmonic coefficients. Compared to the original model, we made four major changes: firstly, we added one layer to each, the generator, the discriminator, and the encoder networks to allow for a higher spatial resolution ($128 \times 128 \times 64$ compared to $64 \times 64 \times 64$ in the original paper). It is a convenient technique, mostly used in GANs and VAEs, to double the resolution in each CNN layer by interpolation. Only in the last layer, we decided to keep the resolution in axial direction constant, in order to keep a tractable size of the data resulting in an anisotropic voxel size. Secondly, we increased the size of the latent dimension to 5000 to account for the higher complexity of the data compared to single volumes. Thirdly, in order to generate anatomically reasonable data, we had to increase the number of channels per layer up to the maximum number which is limited by the GPU memory available. From our experience with the training process, we assume that the use of an even higher number of channels enabled by future developments in hardware, the quality of the FODs would be improved. And, fourthly, we avoided the mode collapse problem, by drastically increasing the contribution of the $L_1$ distance of the loss function by using $\lambda = 50,000$ in Eq. (2), tuning our model more towards a VAE. In previous training attempts, we used $\lambda = 10$ and observed mode collapse. Our trained generator network is able to generate anatomically meaningful FODs as one example of complex, four-dimensional MRI data. However, by visual inspection of the synthetic FODs' single volumes, we found that the generated images appeared moderately noisy and blurry. Such blurriness is a known artefact of VAEs[34,35]. Nevertheless, compared to a pure VAE we tested for FODs with a lower spatial resolution (not shown), our $\alpha$-WGAN generated less blurry FODs with increased spatial details. While brain regions with lower variability in the training data such as the ventricles, basal ganglia or CC are well represented in the synthetic data, the model has demonstrated some difficulties in accurately generating high-variability regions with finely detailed structures, such as cortical regions, characterised by individual configurations of sulci and gyri.

**Fig. 5 | Connectomics.** The figure illustrates an exemplary connectome, derived from a synthetic FOD data set. **A** shows slice views through the whole brain tractogram in axial, coronal and sagittal views. **B** shows the obtained connectome (right), in relation to the whole brain tractogram, shown in a 3D view (left, top row, cf. [A], the AAL parcellation (left, middle row) and the connectome matrix plot (left, bottom row).

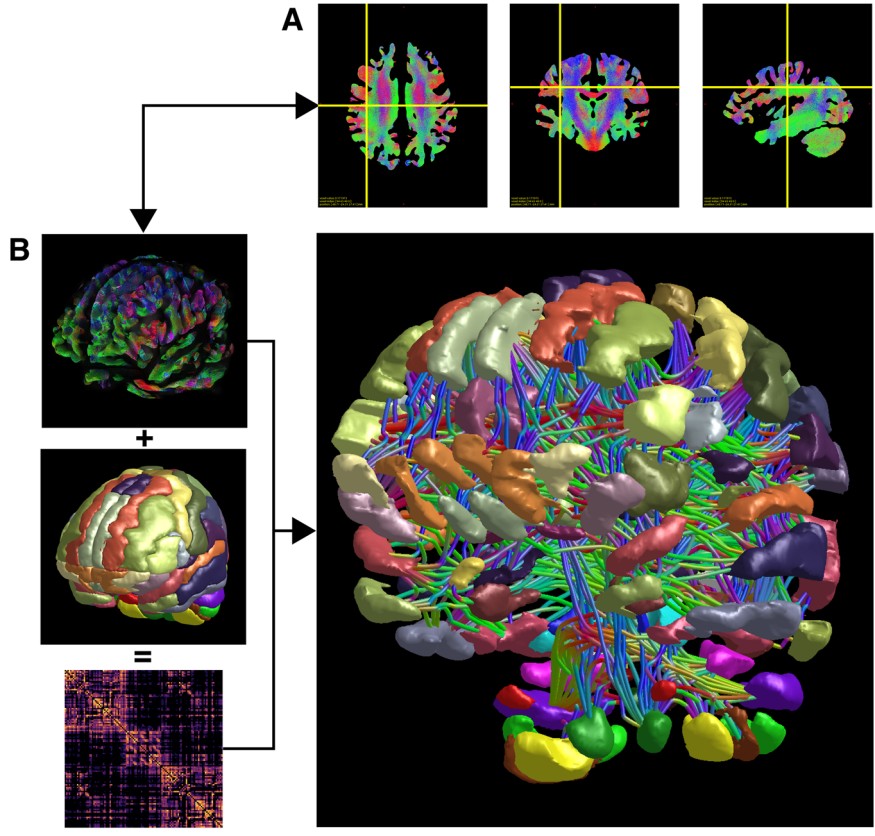

Furthermore, we have shown that our generated data could be processed by common tractography algorithms, yielding reasonable bundle-specific tractograms and even connectomes similar to connectomes derived from real data. We have to mention, that the anisotropic voxel size is non-optimal for tractography in general. However, here we use it to compare our synthetic FODs to real ones regarding their anatomical accuracy, while both types share the same anisotropy. Besides the visual inspection and histogram analyses, we performed the comparison by means of the global efficiency, which is a complex network measure and the Mantel correlation for connectomes, and the dice score, tract volumes and along-tract fibre densities for single tractograms. Global efficiency measures the average inverse shortest path length in a network. It quantifies the efficiency of information exchange over the network. A higher global efficiency means that the brain network can transfer information better between regions. In return, a lower global efficiency indicates a longer average path length, suggesting that the information needs to pass through more steps to go from one region to another. Differences in global efficiency may reflect differences in the structural connectivity between brain regions. A relatively small difference in the mean values. Nevertheless, we expected slight deviations, as the synthesised data may not capture all the properties of the real, structural connections[26]. For the Mantel correlations, we find that synthetic data are slightly less correlated to each other than the validation data which might be a consequence of numerical noise incorporated by the CNNs. As a test of the resulting bundle-specific tractograms, we compared their dice-score distributions. We found that tractograms derived from synthetic data resemble the ones from the validation data, however, the dice scores indicate lower variation in the synthetic tractograms compared to the tractograms derived from the validation data that we already discovered comparing the pairwise mean squared values of each set that is substantially higher on average for the validation data. This observation is in contrast to the mantel correlations, however, it seems that the numerical noise has lower impact on single tracts than it has on whole-brain tractograms, probably due to the restrictions by starting and ending ROIs. Streamlines disturbed by the noise might be more

likely to be rejected, since they do not arrive at the ending ROI while they are counted in the whole-brain tractography. This would also explain the lower volume of the tracts derived from synthetic data. Summing up our analysis, we have shown that our generated FODs capture the essential anatomical features but are still distinguishable from the validation FODs and the variation within the group of generated FODs is slightly smaller.

The main problems using GANs for the generation of complex synthetic data are mode collapse, that we avoided by using the $\alpha$-WGAN and the high demand on computational hardware, especially GPU memory[36]. As a consequence, we downscaled the FODs to keep the generation feasible. The high demand and long training times further complicate detailed optimisation of hyperparameters and network structures, as well as testing of different loss functions such as perceptual loss[37] which might further improve the results. However, novel techniques such as latent diffusion models have shown remarkable results in generating high-resolution $T_2$ volumes[11] and require less computational resources. Such approaches, as well as advances in hardware, might enable us to generate high-resolution FODs indistinguishable from real data in the future. It is worth to note that our approach is not limited to synthetic FODs, but may be generalisable to synthesise any 4D MRI data, such as raw diffusion MR images, functional MRI data or data from an entire MRI protocol containing multiple contrasts. The generation of such multimodal data might be useful in the training of neural networks performing various tasks not only on single volumes, but may take into account multiple modalities or entire MRI acquisition schemes.

The diagnostic by means of clinical images often requires human expertise, interpreting and abstracting to understand disease progression and explain radiological findings. These tasks cannot yet be performed by modern artificial intelligences, especially considering the vast diversity in individual cases. However, AI can automate repetitive tasks such as counting, measuring and segmenting lesions if provided with sufficient amount of data catching the appearing variations in patients[38,39]. Furthermore, with proper training data AI can facilitate the acquisition and processing of complex MRI

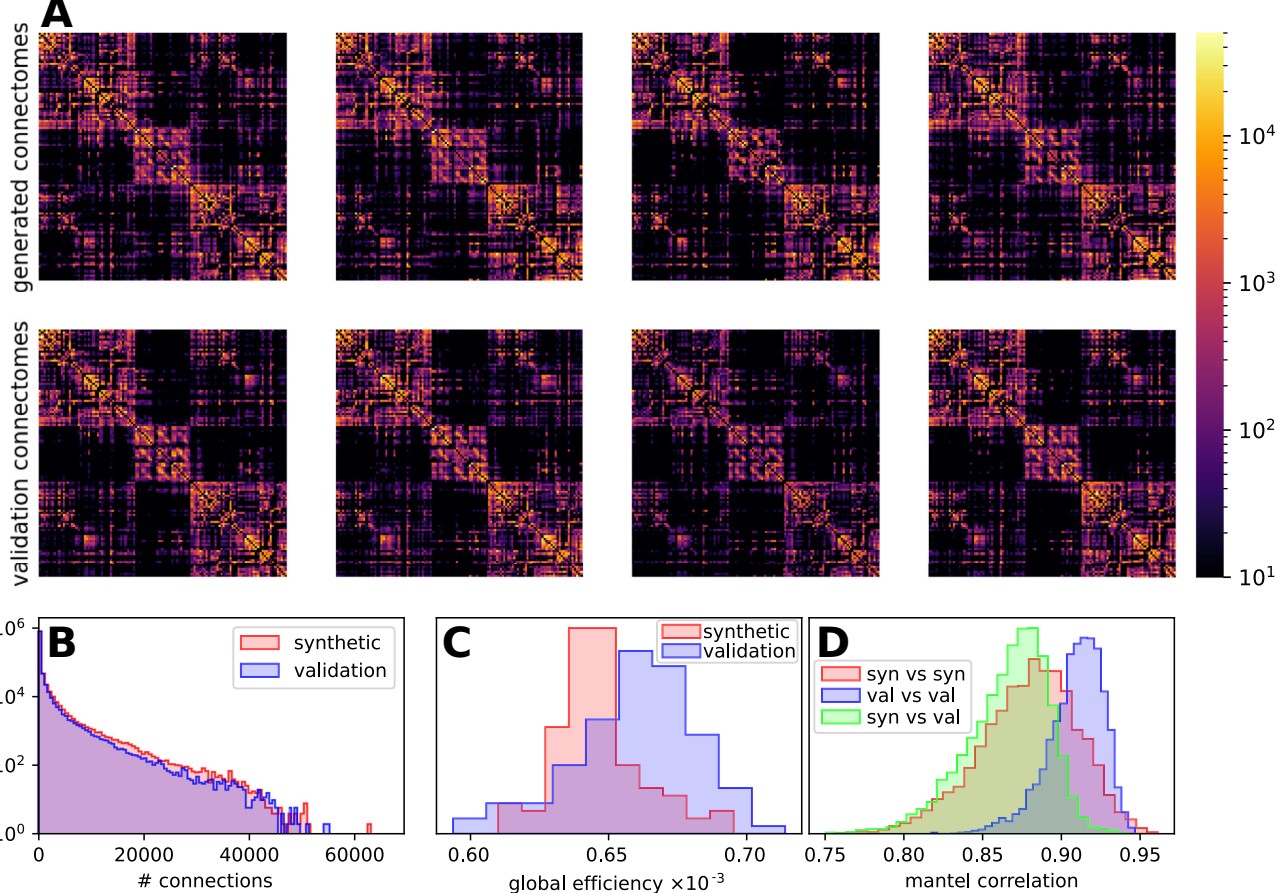

**Fig. 6 | Generated vs. validation connectomes.** We show four randomly selected structural connectomes based on the AAL atlas derived from generated FODs (**A**, top row) and the validation FODs (**A**, bottom row). For comparison, we present the connectome elements of the entire synthetic and validation data as histograms in (**B**), as well as the weighted global efficiency of normalised connectomes in (**C**) and the Mantel correlations in (**D**). Note that the ordinate in panel B is scaled logarithmically while (**B**, **C**) are linearly scaled.

data and, thereby, increasing efficiency which results in better image quality or reduced measurement times[40]. In particular, the acquisition of complex 4D MRI data such as spatially dependent diffusion data or functional MRI is time-consuming and has high demands on hardware. As a consequence, clinical or scientific usage often involves a tradeoff between quality and measurement times, is uneconomically or even impossible. There, the application of AI could allow for a broader application of modern MRI techniques, for instance, introducing connectomes into standard clinical protocols. However, the performance of neural networks on all mentioned tasks depends on the training set that can still be improved by augmentation with synthetic data generated by a GAN. This augmentation can enhance the robustness and generalisability of machine learning models, leading to more accurate and reliable diagnostic or analytical tools[41,42]. This is particularly valuable for rare pathologies, where synthetic data can effectively supplement the limited training data and enhance the performance of neural networks. In this regard, automated tract segmentation tools, such as TractSeg[43], may perform fibre bundle segmentations with high accuracy on healthy subjects. However, a suitable training dataset consisting of pathology-related FOD image acquisitions supplemented by synthetic data could improve performance, such as using data on tractograms of patients with brain tumours. Diffusion MRI GAN may also play a role in mitigating data imbalance, as it can generate additional data for the underrepresented groups, which can lead to more balanced and accurate models. Furthermore, it could also show its value for medical training and the teaching of data analysis, especially if data privacy protection restricts access. One may also think of synthetic MRI sessions serving as a control group for a clinical study with certain age and gender distributions to match a patient cohort or the use of synthetic data for

a preliminary study in which the effect size and, thus, the required number of patients can be estimated. Finally, synthetic FOD data allow for rigorous testing of algorithms and methodologies in a controlled environment, thus leading to the development of more accurate and robust analysis techniques for clinical applications.

## Methods

### Fibre orientation distributions

The diffusion of water in brain tissue provides information about underlying neural microstructure, such as the orientation and density of axonal fibre bundles in the white matter, that can be measured noninvasively along predefined axes by diffusion MRI. A suitable representation for the spatially dependent diffusion within one voxel is the fibre orientation distribution (FOD) that consists of a set of harmonic spherical functions[19,30]. Fitted to diffusion data, the FODs represent a measure of the spatial orientation of axonal fibre bundles and enable us to reconstruct white matter tracts and reveal the brain's structural connectivity[44]. Regarding the FOD representation as sequence of volumes, we follow the conventions from Tournier et al.[30]. In this paper, we consider FODs represented by spherical harmonics up to $l_{max} = 6$ corresponding to 28 three-dimensional volumes per subject.

### Input data, data preprocessing & data preparation

We derived our training data set from the openly accessible data from the from the Human Connectome Project[45] (HCP) S1200 release, available at: https://db.humanconnectome.org. In brief, the HCP is an effort to investigate brain connectivity and function and their variability in healthy adults. The HCP dataset comprises multi-modal imaging, extensive behavioural

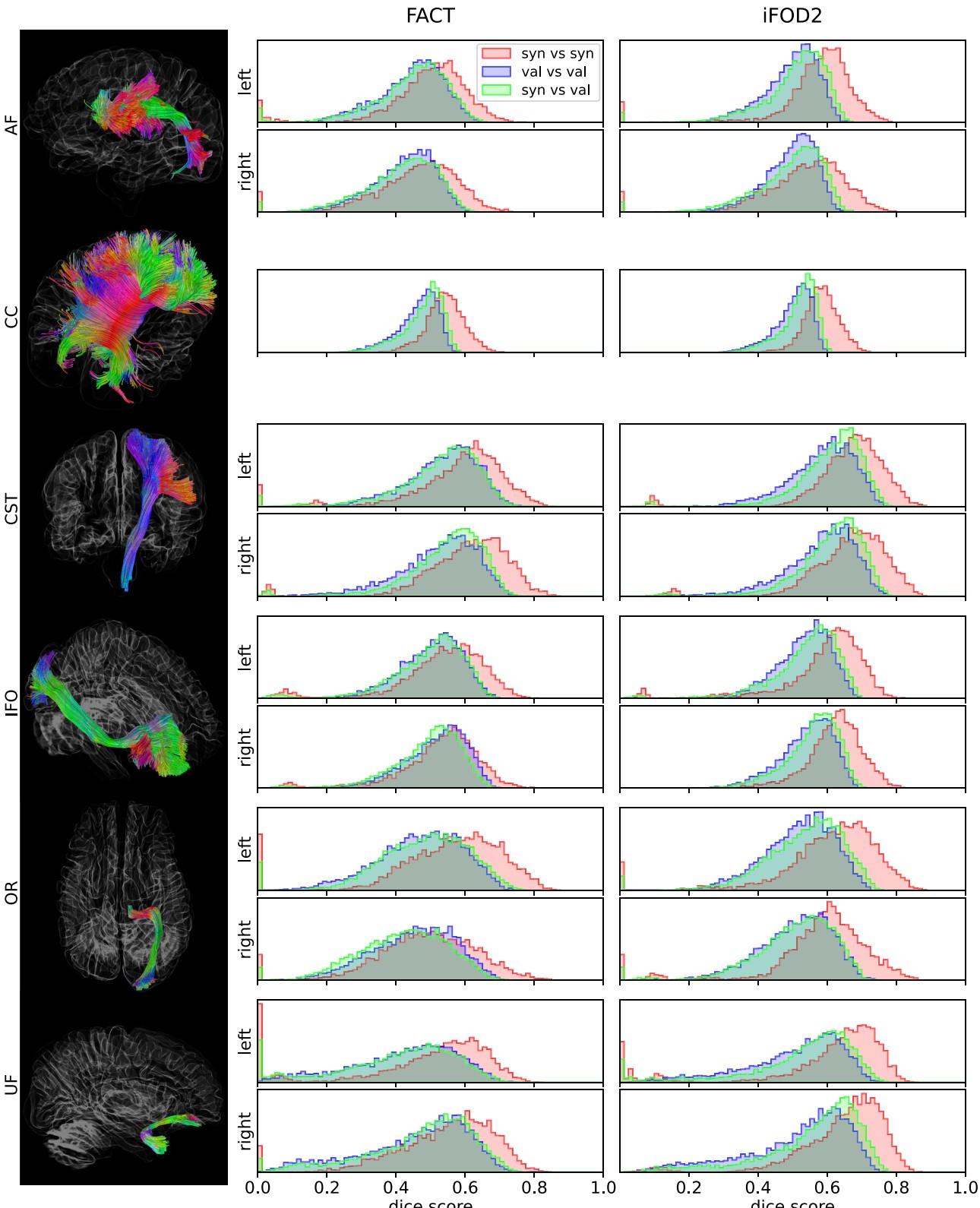

**Fig. 7 | TractSeg results and volume overlap.** The left column shows examples of the tractograms that we derived from one synthetic FOD data set using the FACT algorithm implemented in TractSeg. On the right-hand side, we present histograms of dice scores that measure the volume overlap of selected tracts (arcuate fascicle [AF], corpus callosum [CC], corticospinal tract [CST], Inferior Fronto-Occipital [IFO], Optic Radiation [OR] and Uncinate Fascicle [UF]) of all possible pairs of subjects that are both generated (red), both from the validation set (blue), or one generated vs. one from the validation set (green). We show results from tracts derived with the FACT and the iFOD2 algorithm and on the left and right-hand side, respectively.

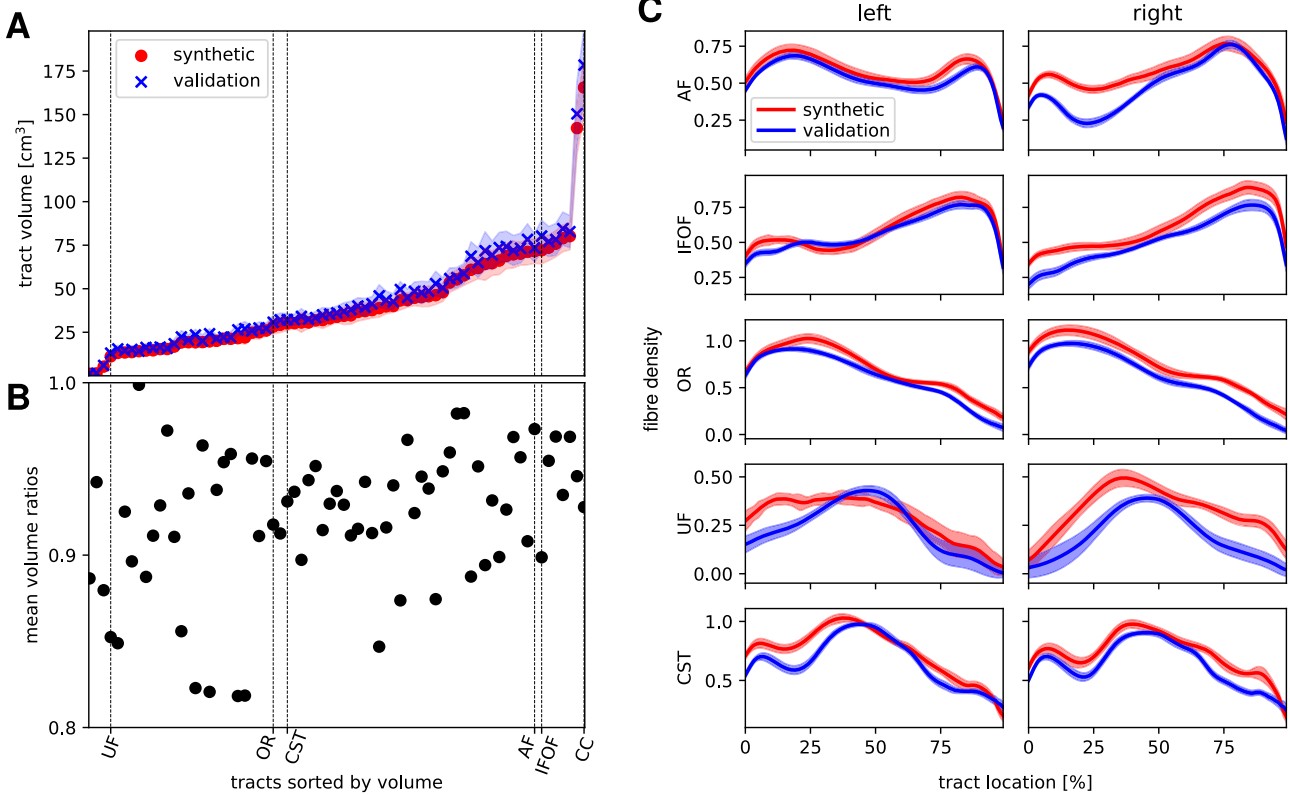

**Fig. 8 | Tract volumes and along-tract fibre densities.** In (**A**), we show the tract volumes from synthetic and validation data starting with the smallest one on the left-hand side to the largest one on the right-hand side. The mean values are presented as crosses and dots, the standard deviation is indicated by the coloured area. In (**B**), we present the mean volume ratio of synthetic to validation data. The dashed vertical lines in (**A**, **B**) indicate the positions of the UF, OR, CST, AF, IFOF and the CC. In panel **C**, we show the fibre density along the indicated tracts. The lines present the mean values at each streamline increment, the areas the standard deviations.

and genetic data from a cohort of 1,065 subjects. MRI data were acquired using a customised Siemens Magnetom Skyra 3T MRI system as part of the Human Connectome Project[45]. For diffusion MRI acquisition, diffusion weightings of b = 0, 1000, 2000 and 3000 s/mm$^2$ were applied in 18, 90, 90 and 90 directions, respectively. In addition, all images were acquired with reversed phase encoding, for the purpose of EPI distortion correction[46]. Other imaging parameters were: TR/TE: 5520/89.5 ms, voxel size: $1.25 \times 1.25 \times 1.25$ mm$^3$, matrix: $145 \times 145$, slices: 174 and NEX: 1. The HCP data are available in a minimally pre-processed form that includes EPI susceptibility-based distortion and motion correction, as well as coregistration of structural and diffusion images[33].

We computed the FOD images using three steps, all performed with the MRtrix3 software package[29,30]. We calculated the response functions by the Dhollander algorithm[47] that are used as Kernels in the second step, where we performed multi-shell multi-tissue constrained spherical deconvolution (MSMT-CSD)[18]. Here, we consider the spherical harmonic functions up to $l_{max} = 6$ which corresponds to 28 coefficients per voxel or, put differently, 28 three-dimensional volumes. $l_{max}$ refers to the maximum spherical harmonic order that we include in the spherical harmonics series. A higher $l_{max}$ allows for a higher level of angular resolution to be represented; however, it also necessitates the storage and processing of more coefficients. Subsequently, in all volumes we removed slices that did not contain parts of the whole brain masks in all subjects and, thus, did not contain relevant information. We kept from the sagittal slices 10 to 137, from the frontal slices 7 to 171 and from the axial slices 2 to 129. Finally, we converted the data to a spatial resolution of $128 \times 128 \times 64$ voxels using MRtrix3[29] resulting in a voxel size of $1.25 \times 1.61 \times 2.5$ mm$^3$. We split the data in a training set containing 965 FOD and a validation set with 100 FOD image data.

We artificially augmented the training data pool by four slightly rotated versions of each data set. We used Gaussian distributed random angles with zero mean and a standard deviation of 1° and spatially rotated the volumes around each main axes. As another augmentation, we also added a version in which we performed three subsequent rotations with independent Gaussian distributed random angles with zero mean and a standard deviation of 1°.

**Model architecture**

To generate synthetic FODs, we adapted the $\alpha$-WGAN (Code available under https://github.com/CUB-IGL/alpha-WGAN-for-FODs) that was initially proposed to generate three-dimensional MRI data by Kwon et al.[23]. The $\alpha$-WGAN model reduces the effect of two issues of classical GANs: mode collapse and instability of the training, by combining the Wasserstein GAN with a variational autoencoder (VAE). The learning process is performed by the interplay of four subnetworks (see Fig. 9): the Generator, the Discriminator, the Encoder and the Code Discriminator.

As in the classical GAN, the Generator transforms a one-dimensional array of Gaussian distributed random numbers $z_r$ into a synthetic data set $x_{syn} = G(z_r)$, here the FODs represented by four dimensional arrays of size $28 \times 128 \times 128 \times 64$. We refer to the number of components of $z_r$ as the latent dimension for which we choose 5000. In the $\alpha$-WGAN it is also part of a VAE in the function of the Decoder. The Generator consists of six layers (see Table 1 for all details). The first layer is a 3D transposed convolutional layer including a batch normalisation and a leaky ReLU activation function. Afterwards, we double the number of increments in each spatial dimension by nearest neighbour interpolation. followed by a convolutional layer, batch normalisation and a ReLU activation function. This block, consisting of interpolation, convolutional layer, batch normalisation and ReLU is repeated four times. After these five blocks, nearest neighbour interpolation is performed only for increments along the frontal and the horizontal axis, followed by a 3D convolutional layer and a tanh activation function. As shown in Table 1, we reduce the output channels from layer to layer.

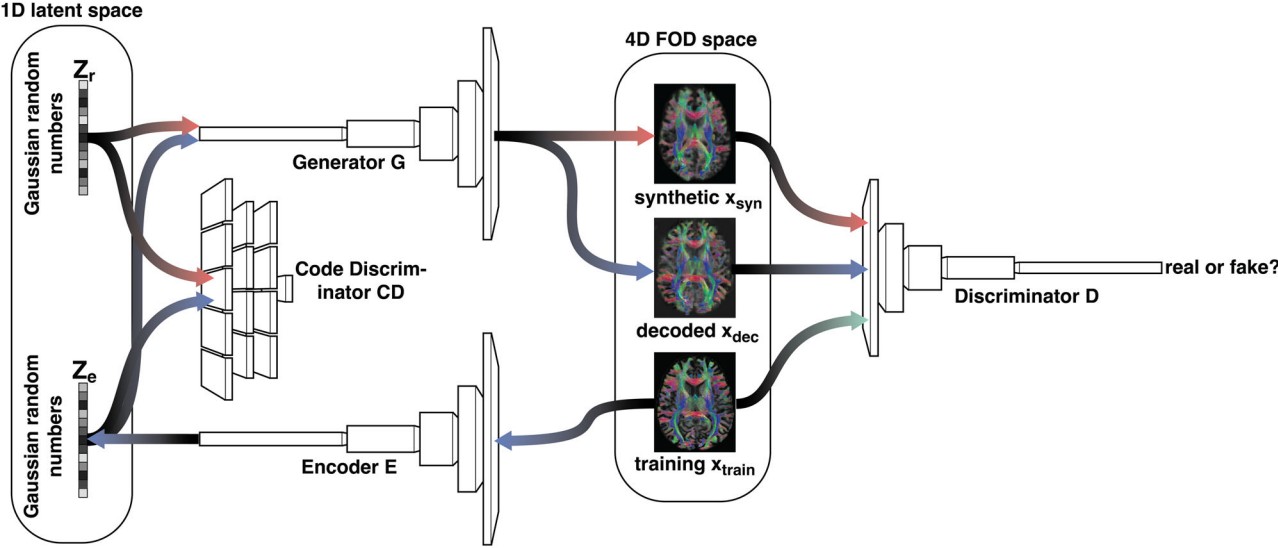

**Fig. 9 | α-Wasserstein GAN.** The network model used in this paper is the combination of the Wasserstein GAN (red path) and a variational autoencoder (blue path).

**Table 1 | Generator implementation details: in the first layer, we apply a 3D transposed convolutional layer and 3D convolutional layers elsewhere**

| Generator layer | scale factor | input channel | output channel | kernel size | stride | padding | batch norm | ReLU neg. slope |
|---|---|---|---|---|---|---|---|---|
| 1 | | 5000 | 3200 | 4 | 1 | 0 | ✓ | 0.1 |
| 2 | 2 | 3200 | 1600 | 3 | 1 | 1 | ✓ | 0.0 |
| 3 | 2 | 1600 | 800 | 3 | 1 | 1 | ✓ | 0.0 |
| 4 | 2 | 800 | 400 | 3 | 1 | 1 | ✓ | 0.0 |
| 5 | 2 | 400 | 200 | 3 | 1 | 1 | ✓ | 0.0 |
| 6 | 2,2,1 | 200 | 28 | 3 | 1 | 1 | | |

In between the layers, we perform nearest neighbour interpolation. Here a scaling factor of 2 means that we double the number of increments in all spatial dimensions and the scale factor 2,2,1 that we double the increments only in the axial layer. After each of the layers 1–5, we use batch normalisations and ReLU activation functions and after the sixth layer, we use tanh as activation function.

The Discriminator is a classifier that aims to distinguish real FODs from the training data from generated FODs. This classification is represented by a single real number output. It consists of six 3D convolutional layers with leaky ReLU activation functions (see Table 2 for all details). After layer 2–5 a batch normalisation is performed.

The Encoder projects the FODs of the training set into a vector in the latent space $z_e = E(x_{\text{train}})$ that has the same dimensionality as $z_r$. Except the output of its last layer, its implementation is same as the Discriminator (see Table 3 for details). Note that the Generator also works as the decoder in a VAE such that $x_{\text{dec}} = G(z_e) \approx x_{\text{train}}$.

The Code Discriminator (CD) classifies data as Gaussian distributed random variables and is trained by $z_r$. It ensures that the encoded training data $z_e$ follow a continuous Gaussian distribution. It consists of three linear layers. After layer one and two, batch normalisations are performed and leaky ReLUs are used as activation functions (see Table 4).

**Model training**

For training, we use Adam optimisation for all networks with a learning rate of 0.0001 and a batch size of 6. We use one batch of generated Gaussian random numbers $z_r$ in the latent space and one batch with randomly selected FODs $x_{\text{train}}$ from the training set. As in the original work from Kwon et al.[23], the first loss function contains the Wasserstein distances used to train the Generator and the Encoder and the $L_1$ distance for FODs from the training data $x_{\text{train}}$ and their corresponding decoded versions

$x_{\text{dec}} = G(E(x_{\text{real}}))$. It reads:

$$L_{G,E} = -\langle D(x_{\text{dec}})\rangle - \langle D(x_{\text{syn}})\rangle - \langle CD(z_e)\rangle + \lambda\langle\|x_{\text{real}} - x_{\text{dec}}\|_{L_1}\rangle, \quad (1)$$

where the angular brackets denote the averages over the batches. Note that the term $\langle CD(z_r)\rangle$ is missing, since both, the generator and the encoder have no effect on it. Compared to the original α WGAN, we drastically increased the influence of the VAE by setting $\lambda = 50,000$ in order to avoid mode collapse of the generated data. To train the Discriminator and the code discriminator, we use the loss functions

$$L_D = \langle D(x_{\text{dec}})\rangle + \langle D(x_{\text{syn}})\rangle - 2\langle D(x_{\text{train}})\rangle + \kappa\phi_D \text{ and } L_{CD} = \langle CD(z_e)\rangle - \langle CD(z_r)\rangle + \kappa\phi_{CD}, \quad (2)$$

respectively, where $\phi_D$ and $\phi_{CD}$ are the gradient penalty terms (see also ref. 48). $\phi_D$ is calculated as

$$\phi_D = \langle(\|\nabla_{\hat{x}}D(\hat{x})\|_{L_2} - 1)^2\rangle + \langle(\|\nabla_{\tilde{x}}D(\tilde{x})\|_{L_2} - 1)^2\rangle, \quad (3)$$

where $\hat{x} = \alpha x_{\text{train}} + (1-\alpha)x_{\text{syn}}$ and $\tilde{x} = \beta x_{\text{train}} + (1-\beta)x_{\text{dec}}$. Again, the angular brackets denote the average over the batch. For each batch component, different random numbers $\alpha$ and $\beta$ are used. $\phi_{CD}$ is calculated similarly: $\phi_{CD} = \langle(\|\nabla_{\hat{z}}CD(\hat{z})\|_{L_2} - 1)^2\rangle$ with $\hat{z} = \gamma z_e + (1-\gamma)z_r$. Here $\gamma$ is a uniformly distributed random number that is drawn for each batch component. As proposed by Kwon et al., we use $\kappa = 10$. Within one epoch,

**Table 2 | Discriminator implementation details: we use six 3D convolutional layers, batch normalisation only in layers 2 to 5 and ReLU activation functions**

| Discriminator layer | inp. channel | out. channel | kernel size | stride | padding | batch norm | ReLu neg. slope |
|---|---|---|---|---|---|---|---|
| 1 | 28 | 125 | 4 | 2 | 1 | | 0.2 |
| 2 | 125 | 250 | 4 | 2 | 1 | ✓ | 0.05 |
| 3 | 250 | 500 | 4 | 2 | 1 | ✓ | 0.05 |
| 4 | 500 | 1000 | 4 | 2 | 1 | ✓ | 0.05 |
| 5 | 1000 | 2000 | 4,4,3 | 2,2,1 | 1 | ✓ | 0.05 |
| 6 | 2000 | 1 | 4 | 1 | 0 | | |

**Table 3 | Encoder implementation details: The encoder has the same structure as the Discriminator, except that the number of output channels in the last layer is 5000 instead of 1**

| Encoder layer | inp. channel | out. channel | kernel size | stride | padding | batch norm | ReLu neg. slope |
|---|---|---|---|---|---|---|---|
| 1 | 28 | 125 | 4 | 2 | 1 | | 0.2 |
| 2 | 125 | 250 | 4 | 2 | 1 | ✓ | 0.05 |
| 3 | 250 | 500 | 4 | 2 | 1 | ✓ | 0.05 |
| 4 | 500 | 1000 | 4 | 2 | 1 | ✓ | 0.05 |
| 5 | 1000 | 2000 | 4,4,3 | 2,2,1 | 1 | ✓ | 0.05 |
| 6 | 2000 | 5000 | 4 | 1 | 0 | | |

**Table 4 | Code Discriminator implementation details: we use three linear layers with batch norm and ReLU activation functions in layers one and two**

| Code Discriminator layer | inp. features | out. features | batch norm | ReLu neg. slope |
|---|---|---|---|---|
| 1 | 5000 | 4096 | ✓ | 0.2 |
| 2 | 4096 | 4096 | ✓ | 0.2 |
| 3 | 4096 | 1 | | – |

we first update $L_{G,E}$ and perform one optimiser step for $E$ and two for $G$. Afterward, we perform four loops of updating $L_D$ and perform an optimiser step for $D$. At the end of each epoch, we update $L_C$ and perform the update step for $CD$. In total, we trained our network for 37,000 epochs with a computation time of seven days.

## Tractography, connectome construction and single tract generation

As an example of usability of the generated data, we reconstructed structural connectomes. To consider all spatial dimensions equally, we regridded the data to an isotropic voxel size of 1.25 mm using MRtrix3. Subsequently, probabilistic tractography was performed with the 2nd-order integration over FODs (iFOD2) algorithm[29]. We set the maximum streamline length to 250 mm. For each tractogram, we computed five million streamlines. We registered the Automated Anatomical Labelling atlas (AAL)[49] to the individual space obtaining subject-specific parcellations. All results were visually inspected before subsequent computations. Streamline weights were assigned by applying the SIFT2 algorithm[50] to the whole-brain tractogram, determining an appropriate cross-sectional area multiplier for each streamline. We obtained structural connectome matrices by mapping the streamlines based on their assignments to the node-wise endpoints defined in the AAL parcellation. This resulted in weighted, undirected networks represented by symmetric $116 \times 116$ adjacency matrices. We observed higher streamline weights for synthetic data yielding a bias in the

connectomes. To correct for that bias, we normalised the connectomes such that the sum of all connections is one or, put differently, the sum of all components of the connectivity matrix is two considering the double occurences in the symmetric matrix. Furthermore, we reconstructed 72 fibre bundles per dataset using TractSeg version 2.8[43]. We used two different algorithms: the deterministic, FOD-peak based tractography algorithm FACT and the FOD-based, probabilistic iFOD2 algorithm[29].

## Validation

For validation, first of all we visually inspected the FOD volumes and compared different contrasts for different anatomical regions. We also compared the FOD values by histogram analysis. Furthermore, we compared two FODs by calculating the sum of voxelwise squared differences. We used the distribution of these squared differences for all pairs within one dataset to quantify its variation. We compared connectomes from the generated data with connectomes from the validation data by means of two measures. Firstly, we compared the global complex network measure global efficiency for weighted networks[26] that we calculated using the brain connectivity toolbox in Python (bctpy 0.6.1). Secondly, we performed the mantel test[28] for the connectomes using the mantel 2.2.0 package in Python. From the single tracts, we considered the CC (commissural fibres), the left and right AF (association fibres) and the left and right corticospinal tracts (projection fibres) as well as the inferior fronto-occipital fascicle, the uncinate fascicle and the optical radiation of both sides. Using MRtrix3, we mapped the selected fibre bundles as binary masks with an isotropic voxel size of 1 mm and compared the volumes as well as the shapes of the resulting tractograms by calculating the Sjørensen-Dice score (cf.[51,52]) for two binary masks **A** and **B** as:

$$S(\mathbf{A}, \mathbf{B}) = \frac{2 \sum \mathbf{A} * \mathbf{B}}{\sum \mathbf{A} + \sum \mathbf{B}}, \qquad (4)$$

where * denotes an element-wise product and the sums go over all three spatial dimensions and, thus, represents the volumes of the bundles. Moreover, we calculated the apparent fibre density from the FODs following the procedure in[53] using MRtrix. We restructured the streamlines of the tracts into 100 equidistant points and sampled the fibre densities along these

tracts. We also tested our data for a regionally specific bias by determining the mean and standard deviations of the fibre densities in AAL parcels. For comparison, we calculated the relative differences in mean ($\overline{fd}$) and standard deviations ($\sigma$), that we define as

$$\Delta\overline{fd} = \frac{|\overline{fd}_{\mathrm{syn}} - \overline{fd}_{\mathrm{val}}|}{\overline{fd}_{\mathrm{syn}} + \overline{fd}_{\mathrm{val}}} \text{ and } \Delta\sigma = \frac{|\sigma_{\mathrm{syn}} - \sigma_{\mathrm{val}}|}{\sigma_{\mathrm{syn}} + \sigma_{\mathrm{val}}}. \quad (5)$$

## Computational resources

We conducted the experiments at our HPC environment of the Charité - Universitätsmedizin Berlin. The model was trained on an NVIDIA DGX A100 80G system, with 128 AMD Epyc cores (Rome), 2TB RAM, 32TB local scratch space and 8x NVIDIA A100 80G GPUs. Both the frontend and the computing nodes run CentOS 8.3. We used Python 3.10 with the Pytorch deep learning library. Training required around 7 days of computation time.

## Reporting summary

Further information on research design is available in the Nature Portfolio Reporting Summary linked to this article.

## Data availability

The code used for initialising and training of the neural networks, as well as the trained neural networks, scripts to generate FODs and used synthetic and validation data are available under Zenodo (https://zenodo.org/records/13902256 version 3) as *synthetic_FODs.zip, validation_FODs.zip, synthetic_connectomes.zip* and *validation_connectomes.zip*. Global efficiencies presented in Fig. 7C are up-loaded as *global_efficiency_validation.txt* and *global_eff iciency_synthetic.txt*, mantel correlations presented in Fig. 7C as *mantel_correlations.txt*. Dice scores presented in Fig. 8 are uploaded as *dice.zip*. Presented and analysed tracts are uploaded as *tracts_synthetic.zip* and *tracts_validation.zip*. Tract volumes presented in Fig. 9A are uploaded as *tract_volumes_synthetic.txt* and *tract_volumes_validation.txt*. Fibre densities measured along the tract as presented in Fig. 9C are uploaded as *along_tract_fd_validation.zip* and *along_tract_fd_validation.zip*. All other data is available on request.

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

## Acknowledgements

L.S.F., S.V. and T.P. acknowledge the support of the Cluster of Excellence Matters of Activity. Image Space Material funded by the Deutsche Forschungsgemeinschaft (DFG, German Research Foundation) under Germany's Excellence Strategy—EXC 2025—390648296. TR received support from the Finnish Cultural Foundation. Funding was provided by Deutsche Forschungsgemeinschaft (DFG, German Research Foundation), SPP 2041, Project number 455227709 and under Germany's Excellence Strategy—EXC 2025—390648296, also funded by the DFG. Dogu Baran Aydogan has received funding from the Academy of Finland (grant no: #348631). The authors acknowledge the Scientific Computing of the IT Division at the Charité—Universitätsmedizin Berlin for providing computational resources that have contributed to the research results reported in this paper. https://www.charite.de/en/research/research_support_services/research_infrastructure/science_it/#c30646061.

## Author contributions
Conceptualisation: L.S.F., S.V., B.A., T.R.; Methodology: S.V., L.S.F.; Investigation: S.V., L.S.F., T.R., B.A., A.C.; Visualisation: S.V., L.S.F., T.R., B.A.; Funding acquisition: T.P., L.S.F.; Project administration: L.S.F.; Supervision: L.S.F.; Writing—original draft: S.V., L.S.F., T.R., B.A., A.C.; Writing—review & editing: L.S.F., S.V., T.R., B.A., A.C., T.P.

## Funding

## Competing interests
The authors declare no competing interests.
