## [Transparent Peer Review file · Communications Biology]

Diffusion MRI GAN synthesizing fibre orientation distribution data using Generative Adversarial Networks

Corresponding Author: Dr Sebastian Vellmer

This manuscript has been previously reviewed at another journal that is not operating a transparent peer review scheme. The manuscript was considered suitable for publication without further review at Communications Biology.

This file contains all author rebuttals in order by version.

Dear Reviewers,

thank you very much for the constructive feedback and reviews. We have revised our paper to address the concerns of the third reviewer. In this letter, we send you our detailed reply to all comments and explain all the changes that we made. The reviewer's comments are indented and italic. We marked all changes in the manuscript in red color. First, we wrote a shorter abstract to fulfill the word-count limit of communications biology. The new abstract reads:

“Machine learning may enhance clinical data analysis but requires large amounts of training data, which are scarce for rare pathologies. While generative neural network models can create realistic synthetic data such as 3D MRI volumes and, thus, augment training datasets, the generation of complex data remains challenging. Fibre orientation distributions (FODs) represent one such complex data type, modelling diffusion as spherical harmonics with stored weights as multiple three-dimensional volumes. We successfully trained an α -WGAN combining a generative adversarial network and a variational autoencoder to generate synthetic FODs, using the Human Connectome Project (HCP) data. Our resulting synthetic FODs produce anatomically accurate fibre bundles and connectomes, with properties matching those from our validation dataset. Our approach extends beyond FODs and could be adapted for generating various types of complex medical imaging data, particularly valuable for augmenting limited clinical datasets.”

Reviewers' comments

Reviewer #2

I have no further comments.

We appreciate Reviewer #2's thorough evaluation and confirmation that our revisions have adequately addressed their previous concerns.

Reviewer #3

The authors present a-WGAN, an improved GAN that is capable of generating complex fiber orientation distribution functions (fODFs) with spherical harmonics of order 6. Such fODFs can represent the anatomy of white matter fibers with high angular resolution and are commonly employed by the HARDI family of diffusion weighted imaging models. The authors base their training on the openly accessible Human Connectome Project dataset. This data is then used to validate the output of the model in various ways. Overall, the model seems to generate realistic fODFs. I appreciate that the authors present several validation analyses, including tractography of single tracts and connectome topology. This is a nice advancement of AI-based approaches in the field of diffusion MRI.

One important finding is that tractography based on the generated fODFs fails to consistently capture certain tracts, e.g., the uncinate fasciculus. This indicates that the model requires substantial improvement before it can reliably generate usable fODFs for broad research applications. An open question on this front is whether the quality of the generated fODFs is consistent in the whole brain or varies systematically. Answering this question is crucial as it affects the type of research projects the model can assist with. E.g., if the reliability of the generated fODFs is lowest in the inferior fronto-temporal region, then the model should not be used to answer questions about tracts that traverse this region. Answering this question may also shed some light on why some tracts show a reduced volume effect more than others. It may very well be that the model is generating less realistic fODFs in regions most highly affected by EPI distortions because the training data there is less reliable. Another shortcoming is highlighted in Fig. 9C, where the along-tract fiber density

of the model-based tracts show clear departures from the empirical ones.

We thank Reviewer #3 for their valuable critics. We agree that for some clinical applications substantial improvements of the model may be required. However, here we provide proof of principle and the important first step, demonstrating that it is already possible to generate anatomically plausible FODs which has not been shown before, at least to the best of our knowledge. In the rapidly evolving field of generative AI, superior methods may have already been developed and an improvement of the neural network would be a consecutive step, as we also mention in the discussion.

The UF could not be reconstructed only 4/100 cases using the FACT algorithm that is known to be unreliable in presence of noise. Using the more robust iFOD2 algorithm, we observe only 2/100 cases in which reconstruction of the UF failed. There, we discovered discontinuities in the bundle segmentation of Tractseg that might be caused by numerical noise in the FODs. Since our synthetic FODs are generated from Gaussian random numbers, we would expect outliers for unlikely input sets that may be difficult to be interpreted by TractSeg. In our paper, we added in lines 246-247:

”However, in two of the generated FODs, small fibre tracts such as the UF could not be reconstructed **even with the iFOD2 algorithm. In both cases, the UF fibre-bundle mask derived with tractSeg was discontinuous.**”

And in the discussion:

The relatively high peaks at zero dice scores in Fig. 8 might be misleading due to the small bin width that we used. The question of FOD reliability in certain regions is indeed highly interesting. To answer this, we use the fibre distributions within the parcels of the AAL atlas for all synthetic and validation FODs, respectively. If we found more than 100 voxels for each group and parcel, we calculated the parcel-wise mean and standard deviation. In Fig. 1, we present the relative differences in mean values and standard deviations, respectively, defined as:

$$\Delta \overline{fd} = \frac{|\overline{fd}_{syn} - \overline{fd}_{val}|}{\overline{fd}_{syn} + \overline{fd}_{val}} \quad (1)$$

$$\Delta \sigma = \frac{|\sigma_{syn} - \sigma_{val}|}{\sigma_{syn} + \sigma_{val}} \quad (2)$$

(Eq. 5 in the new version of our manuscript). Where \overline{fd}_{syn} and \overline{fd}_{val} are the mean fibre densities and σ_{syn} and σ_{val} the standard deviations of synthetic and validation FODs in AAL parcels, respectively. For both mean and standard deviation differences, we only find a clear discrepancy in the cerebellum. For cortical areas, we do not find a region specific bias but only low relative differences. Higher differences in the cerebellar regions might be the result of very high inter-subject variability increasing the learning difficulty. We have added the following sentences in the methods part of our paper in lines 175-176:

“**We also tested our data for a regionally specific bias by determining the mean and standard deviations of the fibre densities in AAL parcels. For comparison, we calculated the relative differences in mean (\overline{fd}) and standard deviations (σ), that we define as ...**”

And in the results part lines 259 to 262:

“The relative differences of fibre-density mean and standard deviations (see Eq. 5) in the AAL parcels reveal a bias in the cerebellum where they are up to 12 and 45 percent, respectively. For other regions, we find lower values under 5 and 20 percent, respectively, that are distributed along the cortex without deducible bias towards a certain area. The highly individual structure of the cerebellum may be more difficult to be learned by our model.”

Fig 1: Region specific relative differences between synthetic and validation FODs of fibre density mean (left) and standard deviation (right).

Furthermore, we are confident that our training data is reliable in certain areas due to EPI distortions. In the HCP dataset the acquisition protocol includes anterior-posterior and posterior-anterior phase-encoding directions to enable optimal distortion correction that was performed with FSL’s TOPUP in addition to eddy current and motion correction. However, we agree that this might be an issue if the neural network is trained on clinical data of lower quality.

Overall, I see the potential of the proposed model for generating realistic fODFs, but I think it is still a long way from achieving a sufficiently high level of realism that would make it useful for clinical applications or, more generally, for non-clinical data augmentation. It is also not clear how well the model could be trained from data of clinical quality compared to those from the HCP.

We thank reviewer #3 again and are grateful that they see potential in our work. We agree that future work and improved algorithms are needed for many clinical applications. However, here we want to emphasize that our work is the first step towards augmentation of diffusion data training sets and provides several tests to ensure a sufficient quality of generated FODs.